

# Monitoring the coastal-offshore water interactions in the Levantine Sea using ocean color and deep supervised learning

Georges Baaklini[1], Julien Brajard[2], Leila Issa[3], Gina Fifani[1], Laurent Mortier[1], and Roy El Hourany[4]

[1]LOCEAN Laboratory, Sorbonne University, UPMC Univ Paris 06 CNRS-IRD-MNHN, 4 place Jussieu, 75005 Paris, France
[2]Nansen Environmental and Remote Sensing Center, Bergen, Norway
[3]Department of Computer Science and Mathematics, Lebanese American University, Beirut, Lebanon
[4]Laboratoire d'Océanologie et de Géosciences, Univ. Littoral Côte d'Opale, Univ. Lille, CNRS, IRD, UMR 8187, LOG, 62930 Wimereux, France

**Correspondence:** Georges Baaklini (georges.baaklini@locean.ipsl.fr)

**Abstract.**

Understanding and tracking the surface circulation of the Levantine Sea presents significant challenges, particularly close to the coast. This difficulty arises due to two main factors: the limited availability of in-situ observations and the increasing inaccuracies in altimetry data close to the coastline. Here, we propose a new approach to monitor the interaction between offshore and coastal waters. In this approach, we develop a pattern detection model using deep learning by training the U-Net model on ocean color data to track the interactions between the coastal and offshore water in the Levantine Sea.

The results showed the presence of notable variations in the behavior of coastal currents as they progress northward beyond 33.8 °E. As these coastal currents become increasingly unstable, they exhibit continuous pinching-off events that are missed by conventional observational tools. These pinching-off events, observed especially along the Lebanese coast, manifest in various patterns evolving simultaneously. Typically, these patterns have a relatively short lifespan of a few weeks, appearing and disappearing rapidly. However, these structures can evolve into larger eddies that endure over four months in some years, especially in the Northern part of the Lebanese coasts. Although these structures could be observed during all the seasons, spring consistently records the lowest activity of these structures. Overall, we showed that the pinching-off events were always observed in the eastern part of the Levantine Sea. On the contrary, in the southern part along the Egyptian coasts, the coastal flow is more stable in the southern region, where these events are less frequently observed, with more than 63% of the total observations not exhibiting any pinching-off events. Moreover, when these events occur in the south, their spatial extent is notably limited.

This research not only sheds light on previously missed (or underestimated) coastal current dynamics in the Levantine Sea but also highlights the crucial need to increase in-situ observations to advance our understanding of this region's complex oceanographic processes.



## 1 Introduction

Understanding the coastal circulation is key information for a wide range of applications as it plays a crucial role in the redistribution of water properties, such as nutrients or chemicals from an anthropogenic source channeled through rivers as well as the transport of nutrient-rich coastal waters into the more oligotrophic open sea (Escudier et al., 2016; Levy and Martin, 2013; Taupier-Letage et al., 2003). In the Levantine Sea, the exchanges between coastal and offshore waters could be intensified by mesoscale structures that can trap, advect, and deviate the coastal flow into further distances offshore. Indeed, the Levantine Sea, characterized by a mesoscale activity that is highly dynamic and evolving in a short time scale (Menna et al., 2012), is described as an anticlockwise flow, circulating along the continental slope and interacting with different meso and submesoscale structures evolving offshore (see figure 1). The offshore mesoscale activity has been studied using different approaches: long-term averaged Mean Dynamic Topography (MDT) (Amitai et al., 2010), current flow kinetic energy variability (Pujol and Larnicol, 2005; Menna et al., 2012), eddies tracking (Mkhinini et al., 2014) and neural network classification (Baaklini et al., 2022). On the other hand, the along-slope coastal circulation has been less explored, and there is still a significant gap in our understanding of the interaction between coastal currents and the dynamical features close to the shoreline. This lack of information is primarily due to the rarity of in-situ observations. Moreover, previous studies have predominantly relied on satellite altimetry observations, that is still error-prone near the coastal areas where the satellite information degrades within 20-50 km from land (Cipollini et al., 2010), and thus, a detailed description of the near-coast features can be missed or biased (Fifani et al., 2021; Baaklini, 2022).

A typical case showing the impact of altimetry inaccuracies on monitoring precisely these dynamical features is presented in figure 2 (right panel). In this example, a spatially extended eddy structure was missed by the altimetry velocity field. In contrast, the ocean color image was capable of accurately delineating the eddy structures (left panel). Indeed high-resolution chlorophyll images can detect even small swirling and filamentary patterns of chlorophyll, hence providing more accurate monitoring of different features (eddies, jets, meanders, etc.) (Sarangi, 2012). Although the Levantine Sea is considered the marine equivalent of a terrestrial desert, the use of ocean color images to track the coastal-offshore interactions is still possible mainly due to its along-slope coastal current that interacts with the discharges of the Nile River (the only main river in the Levantine Sea), which causes the emergence of a hotspot of algal blooming occurring close to the Egyptian coast. Moreover, the coastal circulation carries part of the nutrient-rich water further to the east, extending from the coastal area of Egypt to Lebanon, that could evolve into eddies and filaments frequently observed extending seaward from the coast and interacting with larger basin-wide patterns (Barale et al., 2008).

When having an extensive dataset of high-resolution ocean color images, analyzing different features poses challenges using traditional computer vision techniques. These challenges arise from factors such as high temporal variability, limited information during periods of heavy cloud cover, and the intricate behavior of these oceanic features. The emergence of artificial intelligence has allowed the development of deep learning methods that extract features from input images. In physical oceanography, Image segmentation based on deep learning techniques, such as the Convolutional Neural Network (CNN), was previously applied to predict the monthly sea surface salinity (SSS) (Zhang et al., 2023), and even the sea subsurface





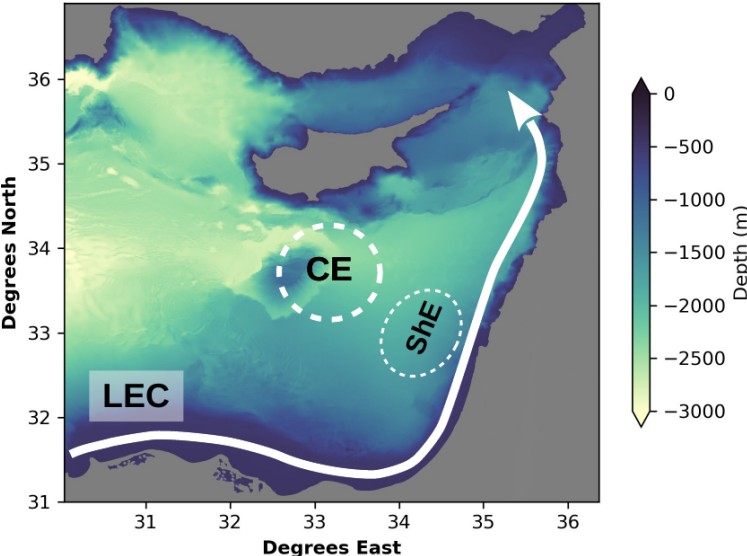

**Figure 1.** A schematic representation of the Levantine Sea surface dynamics in the area from the Nile River discharge until the passage between Cyprus and Syrian coasts. The so-called Libyo-Egyptian along-slope circulation (LEC) is the AW entering the Levantine Sea, flows in a cyclonic along-slope circulation, and interacts with the Shikmona eddy (She), with the presence of a larger mesoscale structure, the Cyprus eddy, located in the south of Cyprus. All is overlayed on a bathymetry map.

temperature (SSbT) (Sun et al., 2022) or for the automated eddy detection and classification from Sea Surface Height (SSH) map (Lguensat et al., 2018), or infrared satellite imagery (Moschos et al., 2023), and even to identify oil spill instances in SAR images (Shaban et al., 2021). U-Net model is a type of the CNN initially developed for the recognition of different cellular structures in biomedical images (Ronneberger et al., 2015) before being applied in several fields of oceanography such as eddies detecting (Moschos et al., 2023) or to track tracers of microborers in coral cores (Alaguarda et al., 2022).

In this paper, we will identify and monitor the coastal-originated water using a deep learning technique (U-Net) and ocean color images to address the following questions: what is the frequency of the coastal-offshore water exchange? In which parts of the Levantine coasts would this eddy formation and pinching-off occur the most? Is there any seasonality in the coastal-offshore interactions? What factors could be influencing these events?

     The paper is structured as follows: In section 2, we provide an overview of the data. Section 3 details the training of the 65  U-Net model. From section 4.1 to 4.3, we present our results regarding the monitoring of pinching-off events. In section 5, we discuss these results before concluding with the main findings in section 6.

## 2   Data

In this section, we present the data and the methodology used to develop the pattern detection model that tracks off-shore coastal-originated water:





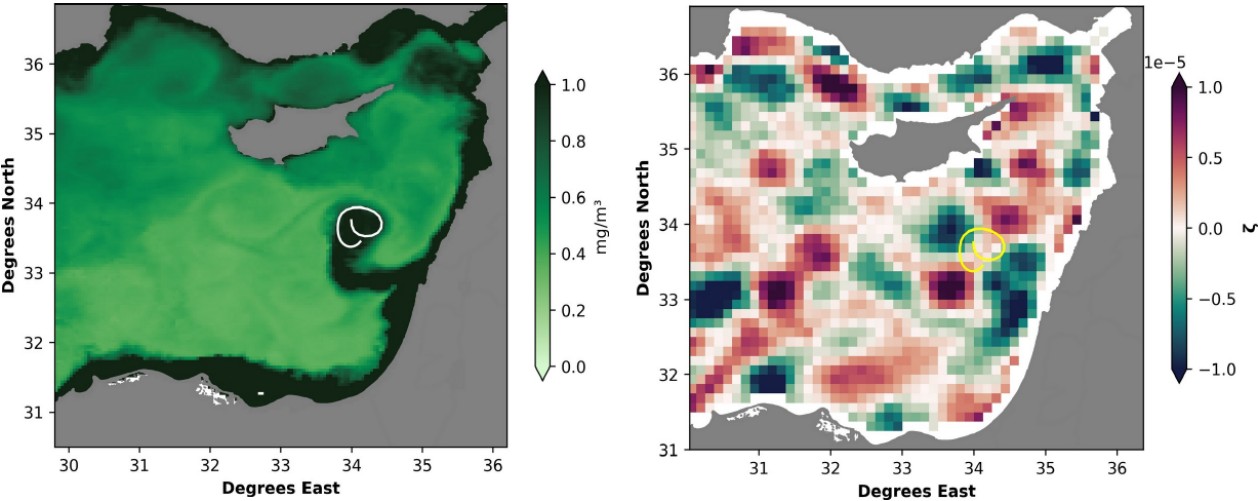

**Figure 2.** The left panel represents the 8-day average chlorophyll image for the days between $21^{st}$ to $28^{th}$ of August 2009, while the figure on the right panel represents the relative vorticity average for the corresponding days computed from the altimetry data. Both are overlaid on the trajectory of a drifter corresponding to these days.

## 2.1 Chlorophyll and Reflectance data

We retrieved MODIS-Aqua chlorophyll (Chla) and remote sensing reflectance at five wavelengths (Rrs at 412, 443, 488, 555, and 678 nm ) level-3 data from GSFC-NASA, with a spatial resolution of 1 km and a temporal resolution of eight days, spanning from 2003 to the end of 2022. The use of Rrs and Chla to detect coastal water advection was inspired by several studies that use these parameters to classify oceanic water masses (Mélin and Vantrepotte, 2015; Spyrakos et al., 2011; Moore et al., 2009; Botha et al., 2020; Martin Traykovski and Sosik, 2003; Jackson et al., 2017; Wei et al., 2022). Spectral classification of satellite Rrs data allows for the distinguishing and grouping of waters with bio-optical and biogeochemical features that may explain the productivity and the constituents of a given water body. Moreover, Rrs provides observations that are independent of the chlorophyll estimation algorithms (Cannizzaro and Carder, 2006). The impact of merging chlorophyll and reflectance data on the model's detection performance is presented in table B2 (in appendix).

Following the exclusion of images under heavy cloud we retained 718 out of the total 920 weekly images per product, which accounts for approximately $80\%$ of the dataset. The temporal distribution of this dataset is visually depicted in figure 3, illustrating that the number of available images varied between 30 and 40 observations per year.

## 3 Method

Detecting and labeling offshore coastal-derived water masses can be challenging. Although the pinched-off water could be visually noticed by ocean color images, using a simple statistical approach to detect it could lead to biases when applied to a larger dataset. A typical case is presented in figure 4, where detecting these relatively chlorophyll-rich waters was based on





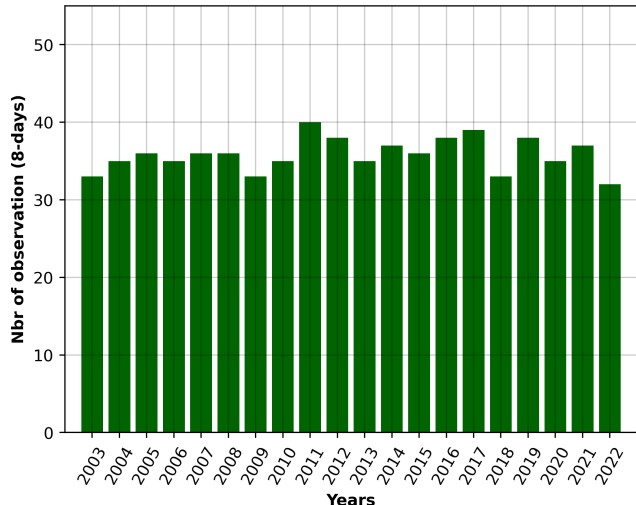

**Figure 3.** The yearly variation of the available ocean color observations (Chla and Rrs) in the Levantine Sea from 2003 to 2022.

values exceeding the top percentile threshold of 20%. This threshold was selected subjectively to optimize the coastal-derived water detection for this case. Although this approach may yield efficient results for the first case (days between $21^{st}$ to $28^{th}$ August 2009, see upper panel), the selected threshold underestimated the density of the pinched-off water for the other case
(days between $26^{th}$ Jun to $3^{rd}$ July 2007, see lower panel). These two examples illustrate that the usage of simple statistical methods might work well for one case but may not necessarily be effective for another. Moreover, such an approach did not accurately delimitate these targeted pinching-off features in both cases. Accordingly, there is a need to develop a more systematic approach to detect pinching-off phenomena.

### 3.1 The U-Net architecture

To achieve our goal, we utilized a Convolutional Neural Network (CNN), specifically the U-Net encoder-decoder architecture, known for precise object detection and identification in image analysis tasks, employing a repetitive process involving consecutive convolutions and downsampling operations (Ronneberger et al., 2015). The final layer applied a sigmoid function (Han and Moraga, 1995) to assign probabilities to pixels, determining the categories based on the highest probability, while training optimized parameters using the Adams algorithm (Kingma and Ba, 2014) and binary cross-entropy loss function.

### 3.2 Input layer

The input layer of U-Net was formed by the reflectance (at wavelengths 412, 443, 490, and 555 nm) and chlorophyll data (see figure 5). The spatial resolution of the input data was downgraded from 1 km to 5 km during the training process. This downsampling was implemented to accommodate computational constraints and to ensure compatibility with the model architecture. To specifically target the coastal circulation, we centered the learning procedure around the Nile and the eastern part





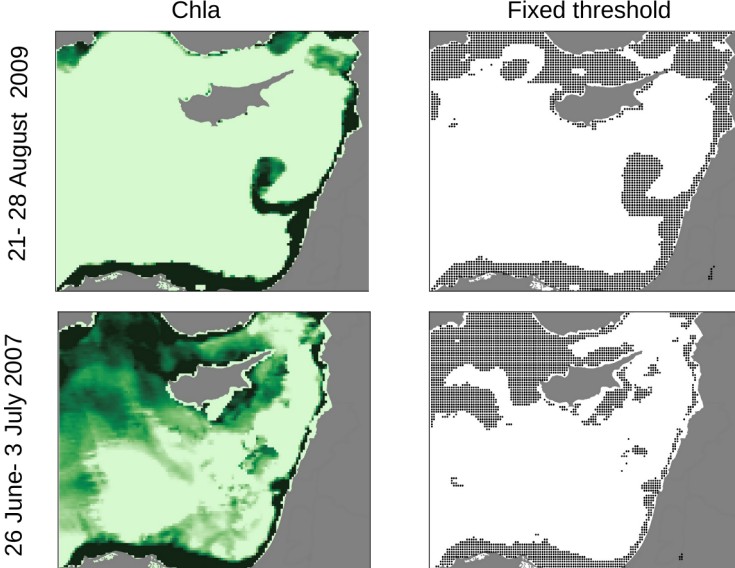

**Figure 4.** The figures in the left panels represent the average chlorophyll images for the days between $21^{st}$ to $28^{th}$ August 2009 and from $26^{th}$ Jun to $3^{rd}$ July 2007. The figures in the right panel show the pinched-off water predicted by a predefined threshold, represented by the dark dots for the two corresponding cases.

of the Levantine Sea. From the total dataset, we selected a subset of 124 chlorophyll images. Each image was divided into the foreground, representing the coastal-derived water (labeled 1), and the background, representing the off-shore water (labeled 0) through a binary classification, with the image manipulation software GIMP (see the middle panel in figure 6). The labeling was performed based on two criteria: the contrast of Chla concentration between the offshore and coastal-derived water (left panel), and the continuity with the coastal Chla gradient. The sharp decrease in the Chla gradient delimits the boundaries
between the coastal-derived features and the offshore water.

The total 124-image dataset was split into a training set consisting of 104 images, a validation set of 10 images for refining hyperparameters, and a separate test set of 10 images to evaluate the final model's performance. To ensure a comprehensive evaluation of our U-Net model's performance by capturing the different environmental dynamics throughout the entire annual cycle, we selected the validation and test datasets in a way that encompasses diverse seasonal variations. To significantly expand
the training set, which is crucial for the machine learning approach, we applied a data augmentation technique by rotating the image left, right, up, and down. This technique allowed to increase the training dataset from 104 to 428 images. Such increase had a significant impact in increasing the accuracy of the model (see table B1) .

Model training allowed to identify coastal-originated water, with its predictions expected to produce values close to 1 for this class, indicating a high probability of coastal water presence. A typical example of the U-Net model detection of the coastal
water is presented in figure 6 (right panel).





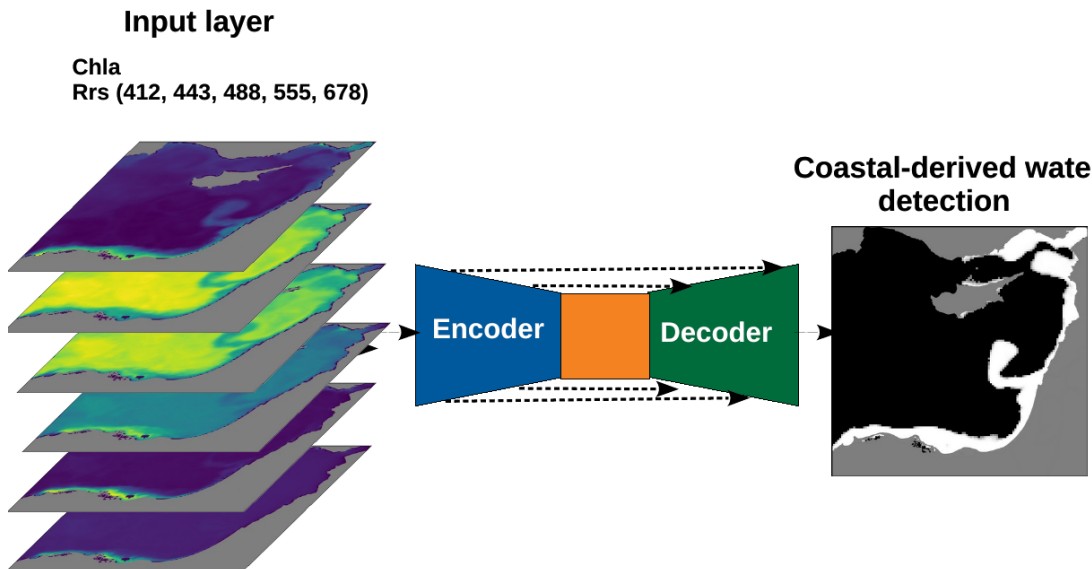

**Figure 5.** The input layer, consisted of Chla (chlorophyll-a) and reflectance images (at wavelengths 412, 443, 488, 555, and 678 nm) that will be down-sampled by the encoder before restoring the dimensions to the original size of the input image by the decoder. The final output layer of the U-Net architecture produces a prediction mask, where each pixel is assigned a class label. The white pixels represent the coastal-derived nutrient-rich water.

## 3.3 Model tuning

Model tuning aims to achieve the optimal model with the lowest loss during the validation. Because the dataset is imbalanced, with more offshore water compared to the coastal-derived, we evaluated the model precision using different metrics such as precision, recall, and F1 score. Moreover, we estimated the percentage of the surface difference between our model predictions and the ground truth data. This percentage represents the variation between the manually labeled surface area and the predictions generated by the U-Net model. More information on the model optimization and fine-tuning scores is presented in table B1 (in appendix).

Furthermore, to assess the model's applicability to new images, we used the test dataset independent of both the training and tuning phases of the U-Net. It serves as a reliable reference to evaluate the model's capacity to generalize its classification performance. The model's performance scored a precision of 0.9, a recall of 0.73, and an F1 score of 0.8, with an average surface error of less than 3 percent when tested on new unseen data.



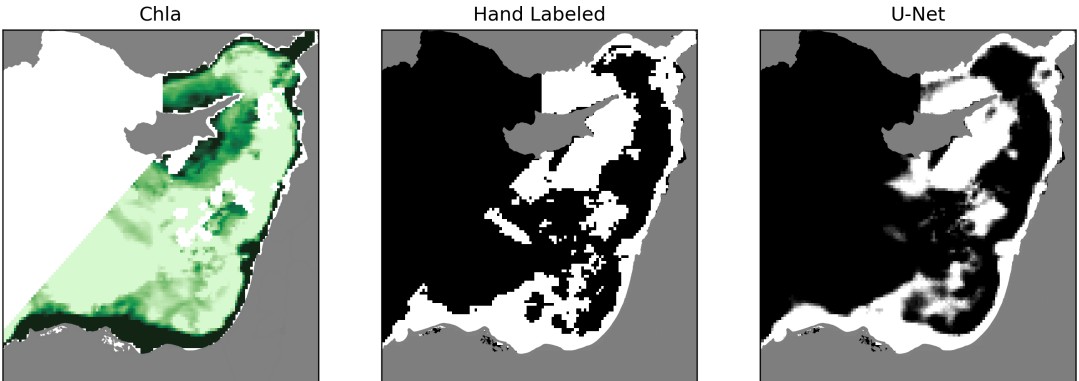

**Figure 6.** The left panel shows the 8-day average Chla images for the days between 28 July $^{th}$ to the $4^{th}$ August 2005. The manually labeled coastal originated water is presented in the middle panel and compared to the result of the U-Net prediction in the right panel.

# 4 Results and discussion

In this section, we will present the results of applying the detection model to the 718 weekly data available from 2003 to 2022:

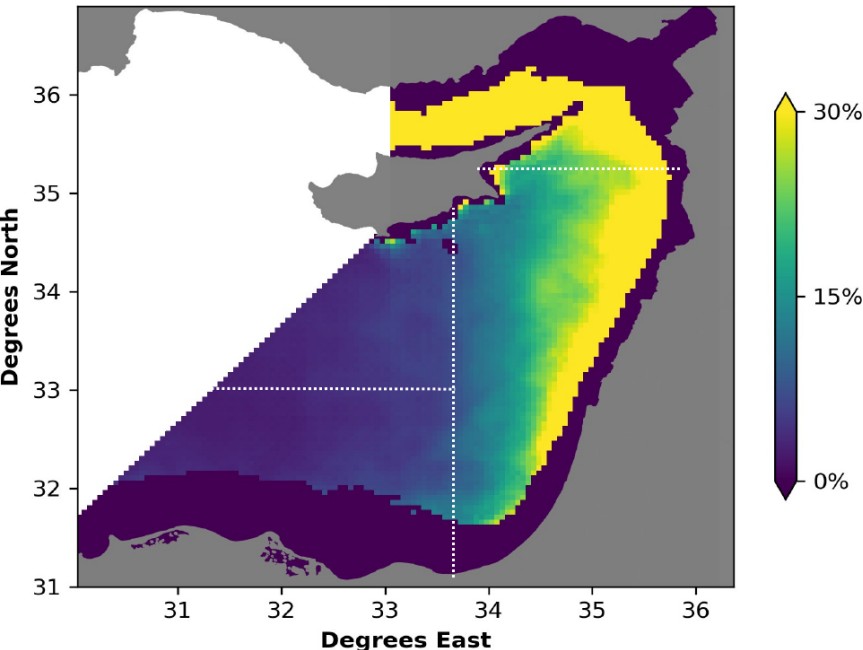

**Figure 7.** The percentage of the coastal-derived water in each pixel of the Levantine Sea averaged from the start of 2003 until the end of 2022. The white vertical lines represent imaginary lines delimiting the south and the east blocks.





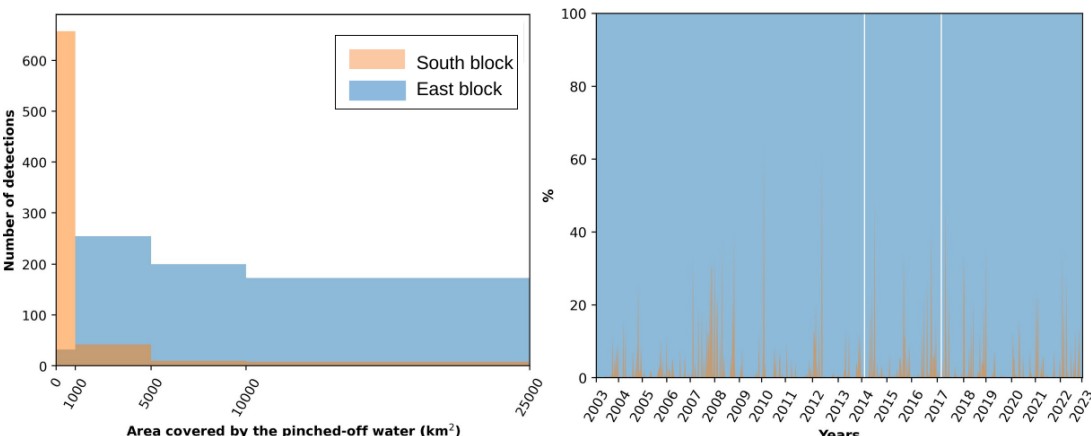

**Figure 8.** The histogram in the left panel illustrates the surface covered by the pinched-off water in the east (blue) and south (orange) blocks. The right panel shows the percentage variation of each block contribution to the total pinched-off water. The white vertical lines represent days where the pinching-off events were absent in both blocks.

## 4.1 Quantification of the pinching-off amplitude

### 4.1.1 Region-based analysis

Figure 7 reveals the spatial distribution of pinched-off water occurrences. The along-slope coastal circulation has been removed from the analysis to isolate and highlight the deviations or pinching-off events.

The results show a clear horizontal gradient existing in the Levantine sea, where the highest frequencies were observed in the easternmost part, starting from 33 °E and up to the north, where in certain areas or pixels within this region, the pinched-off
water was detected more than 30 % of the total observations. On the contrary, in the southern part of the Egyptian coasts, the frequency of the pinching-off events decreases to values that are close to zero.

In order to further evaluate the occurrence of pinched-off water, we divided the basin into two blocks (separated by the dashed line in figure 7). The first block encompasses the region where the coastal flow is mainly eastward extending from the Nile River until 33.8°E. The second block marks the region where the coastal flow shifts northward upon crossing around
145 33.8°E and continues until the passage between Cyprus and the Syrian coasts. In figure 8, the left panel shows the variation of the surface covered by pinched-off water in each of these two blocks. The results reveal that during almost all the studied weeks ($\sim$ 600 weekly observations), the pinched-off coastal water in the southern part did not cover more than 1000 km$^2$, meaning that these events occur in a few parts of this area and fail to evolve into larger features, and remains very limited. On the contrary, in the eastern block, the size of the pinched-off water significantly expands, with coastal water spanning over
150 5000 km$^2$ in more than 400 weekly observations. Furthermore, in over 200 instances of those cases, it extends even further, covering more than 10,000 km$^2$ of the block.





The right panel in figure 8, which represents the weekly percentage contribution of each block to the total pinching-off, shows similar trends. The results reveal that most of the pinched-off water occurs in the eastern rather than the southern part. This dominance is almost permanent but fluctuates interannually. Additionally, it is noteworthy that the southern block experiences a complete absence of pinched-off water, a phenomenon observed in 456 weeks, representing around 63 % of the total studied weeks. On the contrary, pinching-off events are consistently observed in the eastern block.

These results suggest that the coastal flow entering the Levantine Sea starts as a more stable flow while flowing eastward off the Egyptian coasts. Then instabilities increase and the pinching-off events occur mainly when the flow deviates northward around ~34 °E and until the passage between the Syrian and Cypriot coasts. Such a difference could be related to the difference in the bathymetry between these two blocks. Indeed, the Egyptian offshore bathymetry is characterized by a plain extending for several kilometers off the coasts, contrary to the east block where the plain becomes very limited and quasi-nonexistent, which could cause some increases in the vorticity similar to those observed. According to Sutyrin et al. (2009), the deep flow could shift offshore when the bathymetry changes interact with the upper layer to provide an along-slope vortex drift proportional to the basic drift speed and the steepness.

It is noteworthy that although the previous study in Baaklini et al. (2022) highlighted the presence of high vorticity in the coastal current along the Egyptian coast, our findings demonstrate that this elevated vorticity does not lead to the formation of permanent pinching-off eddy events.

### 4.1.2 Yearly comparison

Previous findings have demonstrated that interactions primarily occur within the eastern block, in contrast to the southern region of the Levant, where the pinching-off events are notably limited and entirely absent in half of the observed instances. Consequently, our subsequent sections will be dedicated to an in-depth analysis of the coastal-offshore interactions within the eastern block.

To have a better understanding of the dynamic evolution of coastal water in this block, we present the yearly variation of the pinched-off coastal water density in figure 9. Results reveal significant variations in the spatial extent of pinched-off coastal water from year to year. In certain years, the deviated water may either be limited or fail to develop into stable and enduring features, as observed in 2007, 2010, and 2022. Alternatively, it can evolve into spatially extensive structures, especially notable in 2011, 2012, 2017, and 2020. During these years, clear eddy structures were observed for more than 15 weeks, indicating their persistence for over four months. Overall, these pinched-off structures were primarily located in the northern part of the Lebanese coast, while in the southern part, they are less observed; although there are years where structures evolved close to the coast, such as 2003, 2006, 2013, and 2021, they remain spatially limited.

The results highlight the intricate and stochastic nature of the interaction between offshore and coastal waters. A year characterized by intense pinching-off events and the formation of eddy structures, such as 2017, may be preceded and followed by years with less pronounced pinching-off occurrences. Additionally, the spatial distribution of these structures varies at short-time scale. For instance, a distinct feature could be observed in the northern part of Lebanon, followed by the emergence of eddy structures in the southern region after one year (such as in 2005 and 2006).





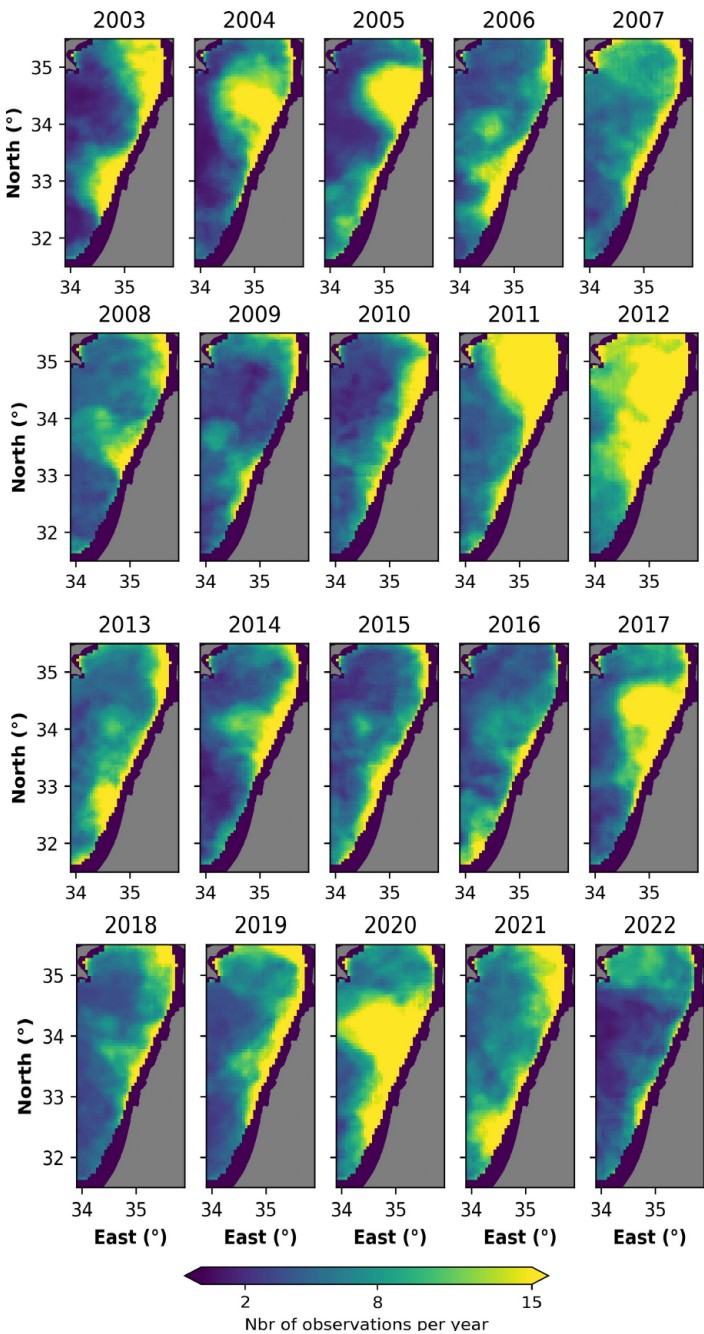

**Figure 9.** The yearly variation of the pinched-off water in the east block from 2003 to 2022. The colorbars indicate the number of observations (based on 8-day images) per year.





While this approach enhanced our comprehension of the coastal current patterns in the eastern part of the Levantine Sea by quantifying the pinching-off events, it did not include an analysis of the evolving dynamical structures. Consequently, during years marked by frequent pinching-off, the understanding of these events remains unclear. For example, the year 2017 shows a prolonged pinching-off event that persisted for over 15 weeks along the coastline of Lebanon. This particular occurrence may be associated with either an elongated and stationary dynamical structure or a structure drifting in proximity continuously close to the coast. To surpass these limitations and better analyze the characteristics of these pinched-off structures, we applied a DBSCAN (Density-Based Spatial Clustering of Applications with Noise) clustering algorithm on the detected patterns. The results are presented in the following section 4.2.

## 4.2 Spatial features within pinched-off structures

The DBSCAN clustering algorithm demonstrates remarkable efficacy in grouping data points that are spatially proximate to each other, using, in our case, the distance criterion derived from the coordinates of pinched-off water. It's important to highlight that, for this approach, we intentionally excluded the along-slope coastal circulation, which is consistently present, in order to focus on the structures of the derived water only.

By employing this approach, it becomes possible to separate distinct features. As an example, in Figure 10, after detecting the pinched-off water in the eastern block of the 8-days averaged observations between $29^{th}$ of August to $5^{th}$ of September 2003, the DBSCAN separates these water into two distinct features characterized by their size and location; the so-called feature 1, which corresponds to an extended eddy located north of the Lebanese coasts, and feature 2, that corresponds to another structure, located more to the south. In this way, due to the size and positioning of features identified using DBSCAN, we are able to distinguish between weeks of a high density of pinched-off water due to extended dynamical structure or to the coexistence of several smaller-scale dynamical features.

We separated these features into three groups based on their sizes: The small-scale; defined as those whose sizes are less than 1000 km$^2$; the mid-scale, which range from 1000 to 5000 km$^2$; and the extended features characterized by sizes exceeding 5000 km$^2$. The percentage variation of the three different groups from 2003 until 2022 is presented in figure 11 (upper panel).

Small-scale and mid-scale features predominated in most years, especially when the pinching-off intensity was weaker (see lower panel), which could probably mean that these structures have a shorter lifetime and/or are not stationary features flowing off-shore along the Levantine slope, failing to evolve into intense and persistent mesoscale features. On the other hand, in the years with the highest persistence, the percentage of extended features increased, surpassing 40% of the total structures, as seen in 2011, 2012, and 2020.

These results reveal a strong correlation between the size of the pinched-off features and the persistence of these events. During the year of the lowest pinching-off events (see figure 11, lower panel), the small-scale structure represented around 50% of the existing features. On the contrary, the highest percentage of extended-scale features with more than 40% observed during years with the highest density of pinched-off water.

Figure 12 illustrates the percentage variations in the number of simultaneous features in the Levantine Sea. It reveals that coastal activity typically involves the coexistence of multiple patterns. The typical scenario involves the simultaneous evolution



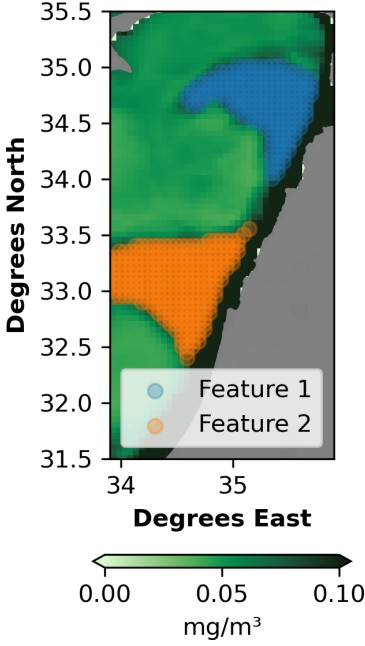

**Figure 10.** An example of the DBSCAN clustering algorithm detecting two distinct features (features 1 and 2) overlaid on the corresponding chlorophyll image taken from August $29^{th}$ to $5^{th}$ of September 2003.

of two to three distinct features, although occasionally, this number may scarcely rise to as many as six evolving structures. However, single-feature dominance was also a common occurrence, and this varied from year to year, with an increase during years when extended features dominated, as seen in 2012.

Overall, these results prove that simultaneous pinching-off events occur at several parts of the coastal flow, leading to numerous features evolving in the eastern block at different spatial scales. These structures are mainly limited spatially and have a short life span, except for the year of intensification. Indeed, the persistence of pinched-off coastal water in the Levantine Sea is directly related to the presence of large-scale structures.

### 4.2.1 Seasonal comparison

In figure 13, we show the annual variation in the total number of pinched-off structures (represented by the dashed lines) alongside the seasonal variations of these events.

The results indicate a general pattern of pinching-off structures evolving across various seasons, with their lowest occurrence within the spring, although there are exceptions in a few years. Moreover, during the years of maximum intensity, with strong and long-lasting structures, such as 2011 and 2012, these events were predominantly observed during the summer and fall seasons. Additionally, it is worth highlighting that the overall count of pinched-off features displays variations from year to year, with the total number of the pinched-off structures fluctuating between $\sim 80$ to $\sim 110$ features per year.





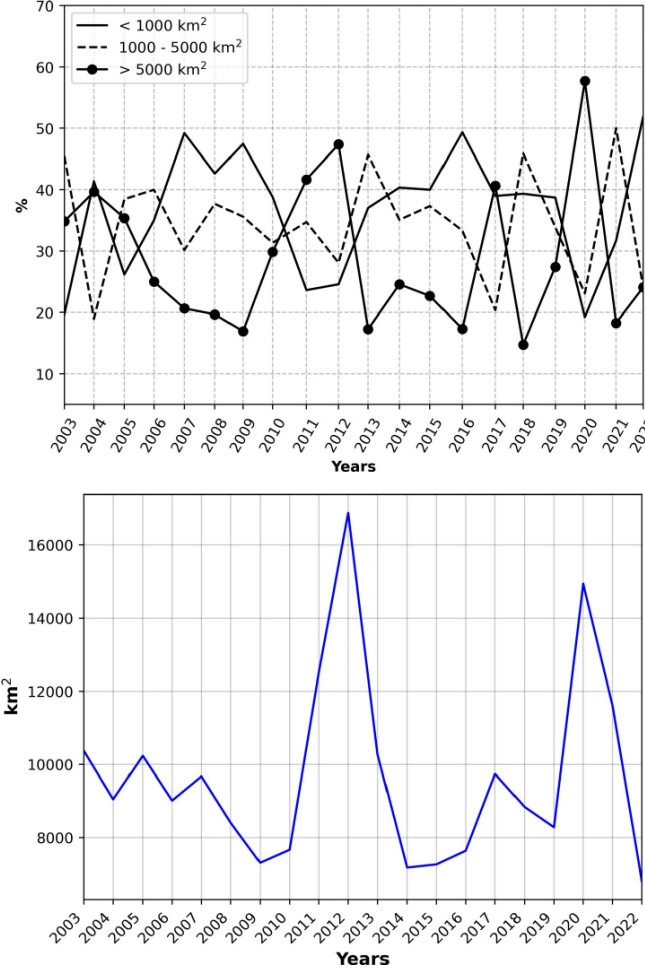

**Figure 11.** The upper panel presents the yearly percentages of small-scale (less than 1000 km$^2$), mid-scale (covering 1000 to 5000 km$^2$), and extended-scale features (more than 5000 km$^2$) in the eastern block of the Levantine Sea from 2003 to 2022. The yearly variation of the mean surface covered by the pinched-off water in km$^2$ is presented in the lower panel.

### 4.3 Comparison with altimetry

The previous findings underscore the complexity of surface current dynamics in the region, which poses challenges for conventional altimetry-based tracking methods. In this part, we compare some of the results obtained with those of the altimetry velocity fields. The surface geostrophic currents are estimated by Optimal Interpolation, merging the measurement from the different altimeter missions available. This product is processed by the DUACS (Data Unification and Altimeter Combination System) multimission altimeter data processing system. It processes data from all altimeter missions: Jason-3, Sentinel-3A, HY-2A, Saral/AltiKa, Cryosat-2, Jason-2, Jason-1, T/P, ENVISAT, GFO, ERS1/2.





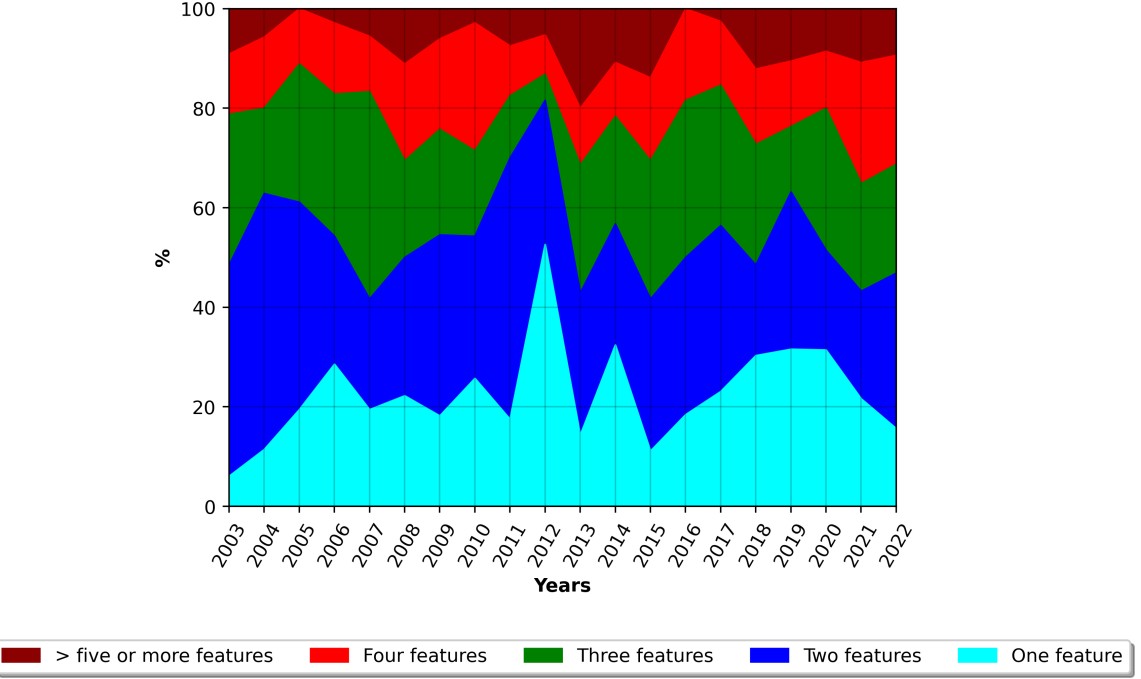

**Figure 12.** The percentage variation in the number of coexisting features observed simultaneously in the eastern part of the Levantine Sea from 2003 to 2022.

Figure 14 shows two cases of 8-day chlorophyll images detecting the same eddy evolving in the north of the Lebanese coast. Despite the eddy's prolonged existence spanning several weeks, the velocity field derived from altimetry data failed to identify a distinct ring-shaped formation when analyzing relative vorticity or the average surface flow. This discrepancy is consistent across multiple years when comparing annual mean velocity fields with the density of the pinching-off events, as shown in figure A1 in the appendix.

From here, although comparing the coastal flow intensity with the variation of the pinching-off events will show no correlation with the pinching-off amplitude, further investigation is needed to assess the impact of intensity on coastal pinching-off events. Indeed, because of the altimetry inaccuracies close to the coast, the accuracy of the surface current representation could lack some precision, and there is a need to have a more precise observation of the coastal current, such as in-situ observation, to better evaluate a potential relation between the flow intensity on the pinching-off events.

## 5 Discussion

The results of monitoring 718 8-day ocean color data in the Levantine Sea from 2003 to 2022 indicated low activity in the southern section of the coastal current (along the Egyptian coast), where pinching-off events were absent in more than half of these observations. In contrast, in the eastern part (along the Lebanese coast), there were notable increases in turbulence,




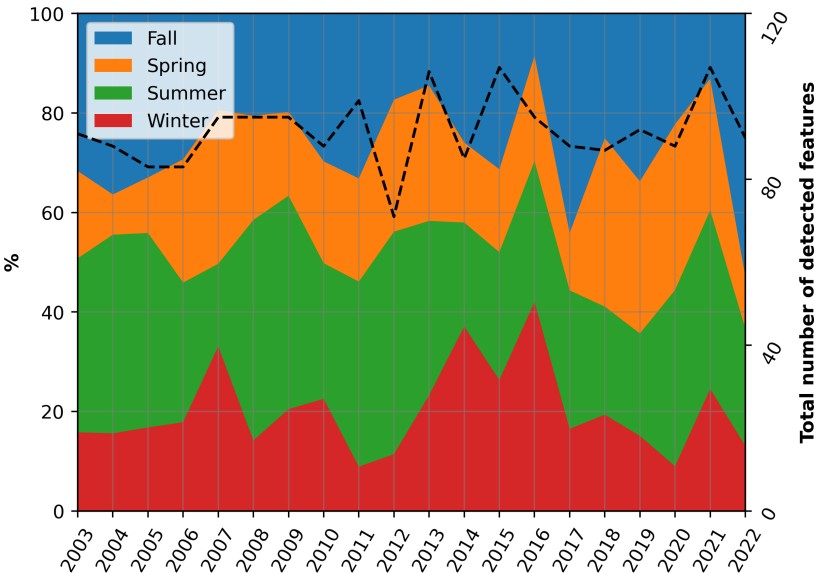

**Figure 13.** The seasonal variation of the pinched-off coastal-derived water in percentage. The dark line represents the variation of the total detected features per year.

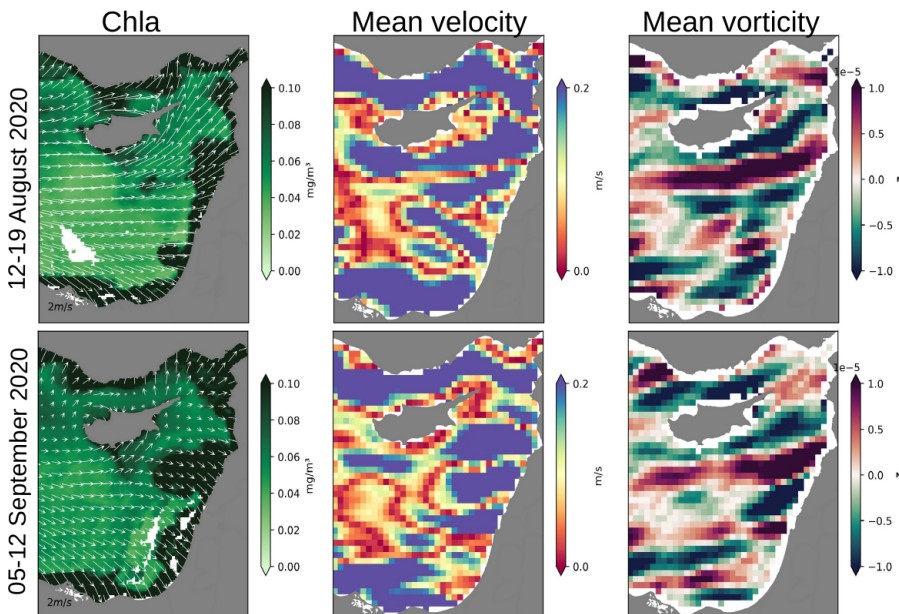

**Figure 14.** Two chlorophyll images (12-19 August 2020 and 05-12 September 2020) (left panel) compared the altimetry-derived average velocity field (middle panel) and relative vorticity (right panel). The chlorophyll images are overlaid on the average velocity field of the average wind speed for the corresponding days.





leading to persistent pinching-off events. Yearly analysis of the pinching-off variation shows that they often evolve into features with relatively short lifespans, lasting only a few weeks and constrained in spatial extent. However, in 2011, 2012, 2017 and 2020 spatially extended and stable structures could be formed, lasting more than four months. The seasonal analysis has revealed that the most pronounced activity of these structures occurs during the summer and fall seasons. Furthermore, while

pinching-off events can be observed throughout all seasons, winter consistently records the lowest activity of these structures.

While pinching-off events are observed along various segments of the Lebanese shoreline, resulting in the concurrent development of multiple evolving features, it is noteworthy that the northern part of the Lebanese coast stands out for its more stable and long-lasting eddies. In contrast, the southern region predominantly reveals the formation of short-lived eddies.

Overall, this complex behavior can be related to variations in bathymetry and convexity of the coastline (Atkinson et al.,

1986; Lillibridge III et al., 1990; Stern and Whitehead, 1990; Ou, 1994). Specifically, a significant transition in bathymetry occurs from a substantial abyssal plain extending several kilometers from the coast to a nearly non-existent plain in the eastern region (approximately around 33.8°E). This shift in bathymetry induces offshore displacement of the deep flow due to interactions between the changing bathymetry and the upper layer, resulting in the generation of an along-slope vortex drift that is proportional to the underlying drift speed and steepness.

In parallel, studies have shown that, in late winter and early spring this coastal current is very weak or completely absent (Rosentraub and Brenner, 2007). However, in summer, this current is well-defined where it separates from the coast and turns westward to the open sea to form large, offshore, cyclonic eddies (Brenner, 2003). In October this current is somewhat weaker than in summer and appears to dissipate to the north. This latter interseasonal variability within the coastal current might explain a part of the seasonal dependence of the pinching-off events.

These findings underscore the dynamic nature of coastal water behavior and offer insights into the diverse range of circulation patterns observed in the Levantine Sea, which cannot be fully captured by current altimetry methods.

## 6 Conclusion

In this study, we explored the coastal dynamics of the Levantine Sea, particularly its interaction with the offshore water. We have developed a pattern detection model that enables the detection and monitoring of derived coastal water deviations from

the coastal flow. The model is based on the U-Net Convolutional Neural Network (CNN) architecture, trained on 8-day high-resolution chlorophyll and reflectance data (wavelengths of 412, 443, 488, 555, and 678). This methodology offers valuable insights into the intricate dynamics between the coast and offshore regions, previously unresolved due to the lack of in-situ observation and altimetry inaccuracies close to the shore, thereby advancing our comprehension of the circulation patterns governing the Levantine Sea.

The reconstruction of the averaged surface current provided by the altimetry did not enable the reproduction of the structures resulting from these coastal offshore interactions. From here, there is a need to increase the deployment of in situ observations in this area. It will provide an accurate assessment of a possible correlation between the intensity of the coastal flow and the





density of pinching off water. For instance, the results of this study second the hypothesis behind pinching-off events appearing to originate from the interaction between the interseasonal variability of the coastal flow and the bathymetric irregularities.

Including SST, or integrating more precise altimetric observations (such as SWOT) might help to better understand these dynamics. Our approach proved to be a promising solution to address the limitations associated with traditional methods of coastal water monitoring and delimitation. Additionally, this method can be expanded to different geographical areas, thereby contributing to an objective characterization of small-scale ocean phenomena. This, in turn, will enhance our understanding of their dynamics and their role in ocean-atmosphere interactions, such as heat and carbon exchange.



**7 Appendix A**

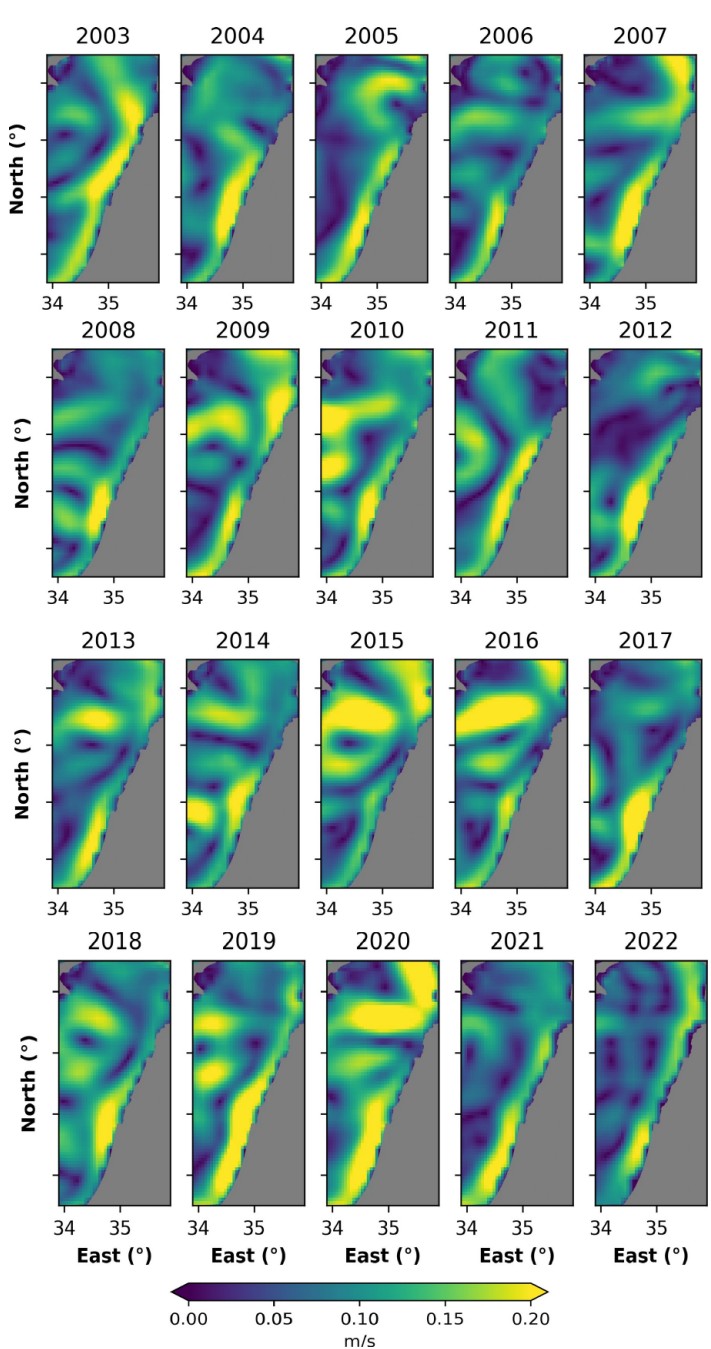

**Figure A1.** The variation of the yearly average velocity field in the east block from 2003 to 2022 (in m/s).



# 8 Appendix B

**Table B1.** Results of the sensitivity test for fine-tuning the U-Net model are presented. The characteristics and scores of the model chosen for this paper's study are highlighted in bold.

| Experiment | Trained data size | Batch size | Nbr of layers | Precision | Recall | F1 score | Avg of wrong surface (%) |
|---|---|---|---|---|---|---|---|
| 1 | 126 | 8 | 4 | 0.98 | 0.34 | 0.52 | -50 |
| 2 | 504 | 8 | 4 | 0.92 | 0.93 | 0.92 | 1 |
| **3** | **504** | **16** | **4** | **0.94** | **0.93** | **0.93** | **-1** |
| 4 | 504 | 8 | 5 | 0.93 | 0.92 | 0.92 | -3 |
| 5 | 504 | 16 | 5 | 0.94 | 0.88 | 0.9 | -5 |

**Table B2.** The impact of changes in the input layer data on the model performance.

| Experiment | Precision | Recall | F1 score | Avg of wrong surface (%) |
|---|---|---|---|---|
| Chlorophyll | 0.91 | 0.93 | 0.92 | 1 |
| Reflectance | 0.94 | 0.88 | 0.91 | -5 |
| Chlorophyll + Reflectance | 0.94 | 0.93 | 0.93 | -1 |

# 9 Code and data availability

The chlorophyll and reflectance data were provided by NASA Goddard Space Flight Center, Ocean Ecology Laboratory, Ocean Biology Processing Group. Moderate-resolution Imaging Spectroradiometer (MODIS) Aqua Chlorophyll Data; 2018 Repro-

300 cessing. NASA OB.DAAC, Greenbelt, MD, USA. doi:data/10.5067/AQUA/MODIS/L3B/CHL/2018. The Bathymetric data used in the figures are GEBCO data with 400 $m$ resolution, available at https://download.gebco.net/.

The Drifters data were provided from: doi:data/10.6092/7a8499bc-c5ee-472c-b8b5-03523d1e73e9.

The altimeter products were produced by Ssalto/Duacs and distributed by AVISO, with support from CNES
(http://www.aviso.altimetry.fr/duacs/).

# 10 Author contribution

The study was conceptualized by GB and REH. The methodology was developed by GB, REH, and JB. Any software used was developed by GB, REH, and GF. Validation was done by GB, REH, JB, LI, GF, and LM. Formal analysis was conducted by GB and REH. Investigation was made by GB, REH, LI, JB, and LM. Resources were obtained by REH and LM. Data curation was done by GB, REH, and GF.



## 11 Competing interests

The authors declare that they have no conflict of interest.

## 12 Acknowledgements

G.B. gratefully acknowledges the Council for Scientific Research of Lebanon (CNRS-L) for their generous support, which has been instrumental in completing GB's PhD thesis, of which this work is a continuation. R.E.H. acknowledges the ANR Chaire Professeur Junior grant number ANR-22-CPJ1-0003-01, and the graduate school IFSEA that benefits from grant number ANR-
21-EXES-0011.



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
