# Peer review of "Monitoring the coastal-offshore water interactions in the Levantine Sea using ocean color and deep supervised learning"

_EGUsphere, 2024_

## Referee Comment (RC1)

The article explores the use of deep supervised learning, specifically a U-Net Convolutional Neural Network, combined with MODIS-Aqua chlorophyll and reflectance data to monitor coastal-offshore water interactions in the Levantine Sea. The study identifies and tracks pinching-off events, where coastal currents form separate water masses, revealing significant variations in these events along the Lebanese coast compared to the more stable flow along the Egyptian coast. Seasonal patterns were observed, with higher activity in summer and fall and lower activity in spring. The study highlights the limitations of traditional altimetry methods in capturing near-coast dynamics and underscores the need for increased in-situ observations and more precise altimetric data. However, I think there are some key problems with this article in terms of modeling, and here are the revisions:

1. Building on the introduction, a brief reference is made to how advances in artificial intelligence and machine learning have revolutionized oceanographic research. This can provide a broader context for the use of deep learning in this study.
2. Measures taken to prevent model overfitting should be discussed in the article to demonstrate model generalizability.
3. Provide more detailed quantitative results on the model's performance, such as precision, recall, and F1 scores, to give a clearer picture of its effectiveness in detecting pinching-off events.
4. Describe the DBSCAN clustering algorithm in detail, discussing the limitations and assumptions of the DBSCAN clustering algorithm and how these limitations and assumptions affect the results. Mention other clustering methods considered and the reasons for choosing DBSCAN.
5. In terms of model selection, I think that U-net alone is not the optimal approach, and improvements to the U-net model should be considered and compared with other semantic segmentation methods, such as Swin-Transformer
6. Add a comparison of the U-Net model's performance with other existing methods for detecting coastal-offshore interactions, highlighting the improvements and any remaining challenges.

---

## Author Comment (AC1)

Dear Reviewer,

We appreciate your valuable and insightful feedback, which improved the manuscript. After carefully considering your comments, we addressed them point-by-point in red, referring to a modified PDF. All the modifications are in red.

1. Building on the introduction, a brief reference is made to how advances in artificial intelligence and machine learning have revolutionized oceanographic research. This can provide a broader context for the use of deep learning in this study.

We have revised the introduction and added references that highlight how AI has significantly contributed to advancing and pushing the boundaries of current knowledge in oceanographic research (50-61)

2. Measures taken to prevent model overfitting should be discussed in the article to demonstrate model generalizability.

Done (118)

3. Provide more detailed quantitative results on the model's performance, such as precision, recall, and F1 scores, to give a clearer picture of its effectiveness in detecting pinching-off events.

The scores are shown in the table of the appendix section

4. Describe the DBSCAN clustering algorithm in detail, discussing the limitations and assumptions of the DBSCAN clustering algorithm and how these limitations and assumptions affect the results. Mention other clustering methods considered and the reasons for choosing DBSCAN.

Done  (211-216)

5. In terms of model selection, I think that U-net alone is not the optimal approach, and improvements to the U-net model should be considered and compared with other semantic segmentation methods, such as Swin-Transformer

Yes, we agree that other methods, such as Swin Transformers, could be more efficient, especially for detecting long-range dependencies in images. However, they tend to be more computationally intensive. Since our main focus in this paper is on local phenomena and considering our computational resources, we opted to use the U-Net, which has proven to be effective for our case and aligns with the aim of the paper.

6. Add a comparison of the U-Net model's performance with other existing methods for detecting coastal-offshore interactions, highlighting the improvements and any remaining challenges.

Done (We added a comparison with another method between 140-145)

---

## Author Comment (AC2)

Dear Reviewer,

We would like to express our gratitude for your valuable reviews. Your insightful feedback has significantly enhanced the manuscript. We have carefully reviewed your comments, shown below in black, and have addressed each point in red. Please refer to the revised PDF, where all changes are in red.

1. I agree with the authors that satellite altimetry is likely to miss small-scale features, especially near the coast but I do not think the right-hand panel in Figure 2 makes the case well. First, the text says "velocity field" while the figure shows vorticity. Second, have the chlorophyll and altimetry fields been closely matched in time? The altimetry field has the correct sized positive and negative vorticity features that, if simply displaced by a mismatch in time, could match up with drifter field (see my comment that a symbol is needed at the start of the drifter track so the reader can tell the direction of flow). Please clear up the use of the altimetry data.
   Done ( please see line 39)

2. More needs to be said about choosing the region of the Nile and eastern part of the Levantine Sea to train the learning procedure. This region should be indicated on one of the maps. What are the consequences if a larger or different region are chosen for training? What biases or errors might be introduced by the choice of training region?
   Done ( please see lines 104-107)

3. Lines 130-131 present the model performance. Please state whether these are "good" scores, perhaps by giving typical values for a good performance from other studies. The 3% error is indeed impressive.
   Done (140-145)

4. Section 4, figure 7: how is the dark blue mask along the coastline chosen? It evidently masks out the coastal water that is on the continental shelf (inshore of some isobath? 200 m?), so that only the coastal water making it into the deep ocean is highlighted in color. Does the sentence "The along-slope coastal circulation has been removed from the analysis to isolate and highlight the deviations or pinching-off events" have something to do with the mask?

   Yes, the sentence "The along-slope coastal circulation has been removed from the analysis to isolate and highlight the deviations or pinching-off events" is indeed related to the mask shown in Figure 7. We removed all the pixels where the detected water occurred more than 60% of the time, which corresponds to quasi-permanent water associated with the continental shelf. We added a sentence to better explain it (in the lines 148-149)

5. In the paragraph about potential instability of the along-coast flow (lines 157-164), a simple explanation is that when the coastal current is over sloping bottom topography the flow is more stable as water parcels tend to follow isobaths. So downstream of where the continental shelf narrows (the start of the eastern block), more of the flow is over the slope and stratification isolates it from the stabilizing influence of bottom topography. Offshore flow can also be caused by flow-topography interaction. I cannot tell from Figure 1 if there are coastline or bottom topographic features that might promote offshore flow. The authors should comment about this possibility.

Done please see 178-181

6. Section 4.2 about spatial scales and temporal persistence is interesting. Regarding the spatial scale, how many eddies based on an estimate of the local internal Rossby radius of deformation can "fit" along the coast in the eastern block? This would help explain how many different structures might be observed.

The coast of the eastern block is approximately 500 km long, while the local Rossby radius of deformation is around 25 km. Given this, we can estimate that around 20 eddies (500 km / 25 km) could theoretically "fit" along the coast. This suggests that multiple eddies, each with a diameter comparable to the Rossby radius, could be present simultaneously along the coastline. However, such estimation remains challenging due to the fact that these eddies can vary in size and structure, and their interactions could contribute to the complex circulation patterns.

7. I am not sure of the purpose of section 4.3 and suggest it could be omitted.
   We agree so we removed it.

Minor/editorial comments:

1. Figure 1 caption: define AW; ShE not She
   Done
2. Figure 2: put a mark on the start location of the drifter
   Done
3. Figure 8 caption: "… (orange) blocks from 2003 to 2023."
   Done
4. Figure 13: how was the top-to-bottom order of the seasons chosen? Not by total, since spring is the least of all. Why not order them by season: fall, winter, spring, summer?
   Done
5. Figure 14: "… chlorophyll images are overlaid by the average …"
   Done

**Citation**: https://doi.org/10.5194/egusphere-2024-1168-RC2